# The Long-Term Evolutionary History of Gradual Reduction of CpG Dinucleotides in the SARS-CoV-2 Lineage

**DOI:** 10.3390/biology10010052

**Published:** 2021-01-12

**Authors:** Sankar Subramanian

**Affiliations:** GeneCology Centre, School of Science and Engineering, University of the Sunshine Coast, Moreton Bay, QLD 4502, Australia; ssankara@usc.edu.au

**Keywords:** CpG dinucleotide, COVID-19, SARS-CoV-2, host defense, virus evolution, adaptation, zinc finger protein, Sarbecovirus

## Abstract

**Simple Summary:**

Severe acute respiratory syndrome coronavirus 2 (SARS-CoV-2) caused the coronavirus disease 2019 (COVID-19), a pandemic that infected over 81 million people worldwide. This has led the scientific community to characterize the genome of this virus, including its nucleotide composition. Investigation of the dinucleotide frequency revealed that the proportion of CG dinucleotides (CpG) is highly reduced in the viral genomes. Since CpG dinucleotides is the target site for the host antiviral zinc finger protein, it has been suggested that the reduction in the proportion of CpG is the viral response to escape from the host defense machinery. In the present study, we investigated the time of origin of reduction in the CpG content. Whole genome analyses based on all representative viral genomes of the group *Betacoronavirus* revealed that the CpG content in the lineage of SARS-CoV-2 has been progressively declining over the past 1213 years. The depletion of CpG was found to occur at neutral—as well as selectively constrained—positions of the viral genomes.

**Abstract:**

Recent studies suggested that the fraction of CG dinucleotides (CpG) is severely reduced in the genome of severe acute respiratory syndrome coronavirus 2 (SARS-CoV-2). The CpG deficiency was predicted to be the adaptive response of the virus to evade degradation of the viral RNA by the antiviral zinc finger protein that specifically binds to CpG nucleotides. By comparing all representative genomes belonging to the genus *Betacoronavirus,* this study examined the potential time of origin of CpG depletion. The results of this investigation revealed a highly significant correlation between the proportions of CpG nucleotide (CpG content) of the betacoronavirus species and their times of divergence from SARS-CoV-2. Species that are distantly related to SARS-CoV-2 had much higher CpG contents than that of SARS-CoV-2. Conversely, closely related species had low CpG contents that are similar to or slightly higher than that of SARS-CoV-2. These results suggest a systematic and continuous reduction in the CpG content in the SARS-CoV-2 lineage that might have started since the *Sarbecovirus* + *Hibecovirus* clade separated from *Nobecovirus*, which was estimated to be 1213 years ago. This depletion was not found to be mediated by the GC contents of the genomes. Our results also showed that the depletion of CpG occurred at neutral positions of the genome as well as those under selection. The latter is evident from the progressive reduction in the proportion of arginine amino acid (coded by CpG dinucleotides) in the SARS-CoV-2 lineage over time. The results of this study suggest that shedding CpG nucleotides from their genome is a continuing process in this viral lineage, potentially to escape from their host defense mechanisms.

## 1. Introduction

Severe acute respiratory syndrome coronavirus 2 (SARS-CoV-2), which belongs to *Betacoronaviruses*, resulted in a pandemic that inflicted respiratory illness in human populations around the globe [1,2,3,4,5,6,7,8]. The outbreak caused by SARS-CoV-2 resulted in >81 million cases and >1.7 million deaths to date. A number of previous studies have characterized the nucleotide composition of the genome of SARS-CoV-2 [9,10,11,12,13,14,15,16]. These studies observed a much high proportion of U nucleotides and concomitant low fraction of C nucleotides in the genome [12,14,15]. The reason for such a discrepancy was attributed to a higher rate of C→U mutations compared to other mutations in the genome. Furthermore, the enzyme APOBEC3G in the viral hosts is known to deaminate cytosine to uracil nucleotides, particularly those present in the single-stranded RNA [14,15,17]. Investigating the dinucleotide patterns of SARS-CoV-2 genomes revealed that the CpG dinucleotides were severely reduced in these viral genomes [14,17,18,19,20]. Zinc finger antiviral protein (ZAP) protects cells from viral infections and this protein identifies viral genomes based on their CpG dinucleotide content (CpG content) [21,22,23]. Therefore, it has been suggested that the low CpG content of SARS-CoV-2 could have evolved to evade identification by ZAP [17].

Although previous studies have characterized the base composition of various viruses belonging to *Betacoronaviruses*, the pattern of evolution of CpG content in the SARS-CoV-2 genomes is unknown. For instance, it is unclear when CpG nucleotides started declining in the lineage of this SARS-CoV-2 and reached the level that we observe in their genomes today. This is important because understanding the evolutionary trajectory of base composition will help predict the future patterns of evolutionary changes. This will eventually assist in designing vaccines that will be effective, not only on the current strains, but also those that evolve in the immediate future. Hence, in the present study, we performed comparative genomic analyses using all representative genomes belonging to the genus *Betacoronavirus* to determine the pattern of evolution of CpG content over time.

## 2. Materials and Methods

### 2.1. Genome Data

Using the Taxonomy Browser application of the National Center for Biotechnology Information (NCBI), GenBank (https://www.ncbi.nlm.nih.gov/Taxonomy/Browser/wwwtax.cgi), we downloaded the complete genome sequences of belonging to the genus *Betacoronavirus*. We selected all representative genomes from the species belonging to the subgenus *Embecovirus*, *Hibecovirus*, *Merbecovirus,* and *Sarbecovirus*. This resulted in 79 complete betacoronavirus genomes, including those infecting human, bat, cow, civet, camel, dog, pangolins, pig, rabbit, rat, and yak (Appendix A). Using GenBank annotations, coding sequence (CDS) regions were extracted for the protein coding genes. Since all mutations at third codon positions of eight amino acids (alanine, arginine, glycine, leucine, proline, serine, threonine, and valine) do not change, the amino acids coded by the respective codons, these positions are considered as synonymous positions. For nonsynonymous positions, we extracted all second codon positions as any mutation in this position changes the amino acid coded by the codon. Excluding the codons UUA, UUG, CUA, CUG, CGA, CGG, AGA and AGG, first codon positions from all other codons are also extracted and considered as nonsynonymous positions. Mutation in the first codon position of the eight codons listed above does not change the amino acid coded by them and, hence, they were not considered as nonsynonymous positions.

### 2.2. Phylogenetic Analysis

We used the Maximum Likelihood (ML) method to infer the phylogenetic relationship among the viral genomes. This was accomplished by using the RaxML program [24]. The sophisticated General Time Reversible (GTR) method was used to model the evolution of nucleotides, as it addresses the differential rates of nucleotide changes, including transition/transversional and base compositional biases. To accommodate rate variations among nucleotide sites, we used the gamma model of evolution. To test the strength of phylogenetic relationships we used the bootstrap resampling procedure with 500 replications.

To compute evolutionary divergence, we first estimated the shape of the rate variation among site (gamma) using the program MEGA [25]. For this purpose, the ML method was used. The HKY model of nucleotide evolution and the discrete gamma model with five categories were used to compute the gamma value. This analysis produced a gamma value of 0.54, which was used to estimate the pairwise evolutionary divergence using the Maximum Composite Likelihood method. To estimate the time of divergence, the Bayesian MCMC method-based software BEAST was used [26]. We opted for the HKY Gamma model for sequence evolution and Yule method for speciation. For calibrating and creating a linearized time tree, a clock rate of 1.1 × 10^−3^ subs/site/year was used [27]. The program was run for 10 million MCMC generations by sampling every 1000 generations. This ensured an Effective Sample Size (ESS) of >200 for all parameters after excluding a 10% burn-in (estimated by the program Tracer [28]). The results were combined from the sampled trees using the program *TreeAnnotator* (supplied as a part of the BEAST software) and the maximum-clade-credibility tree along with divergence time estimates were visualized and drawn using the program *FigTree* (http://tree.bio.ed.ac.uk/software/figtree) (Appendix A).

### 2.3. Estimation of CpG Content

The CpG contents were estimated by counting of the number of CG dinucleotides in a genome and divided by the total number of nucleotides.
(1)$$\mathrm{CpG}\;\mathrm{content}\;\left(\%\right)=\frac{\mathrm{Number}\;\mathrm{of}\;\mathrm{CpG}\;\mathrm{nucleotides}}{\mathrm{Total}\;\mathrm{number}\;\mathrm{of}\;\mathrm{nucleotides}}\times 100$$

To estimate CpG content at synonymous positions we included C in the third codon position followed by G in the (next) first codon position. Similarly, to calculate CpG content for nonsynonymous sites, C in the first codon position followed by G in the second codon position, and C in the second codon position, and G in the third codon position were included. To calculate the ratio of observed to expected CpG content (*I_CpG_*), we used the proportion of CG dinucleotides (*P_CpG_*) in the genome divided by the frequency of C (*P_C_*) and G (*P_G_*) nucleotides [17] as:(2)$$I_{CpG}=\frac{P_{CpG}}{P_{C}P_{G}}$$

## 3. Results

### 3.1. Progressive Decline of CpG Dinucleotides in the SARS-CoV-2 Lineage

The whole genome of SARS-CoV-2 contains 1.47% of CpG dinucleotides, which is much less than that observed for other betacoronaviruses. In order to investigate the time of origin of CpG depletion in the SARS-CoV-2 lineage, we obtained the complete genome sequences of the representatives of all subspecies included in the genus *Betacoronavirus*. The 79 viral genomes (Appendix A) were aligned, and a Maximum Likelihood tree was constructed (Figure 1). Each node of the tree was painted with a distinct color to highlight eight distinct phyletic groups. The CpG content was estimated for each genome and the average CpG content was estimated for each phyletic group. As shown in Figure 1—inset and Figure 2, the CpG content is the highest for the *Nobecovirus* genomes (red circles), which progressively declines towards the descending phyletic groups and reaches to the lowest for SARS-CoV-2 and its close relatives (dark blue circles). In contrast, the ancestral groups *Merbecovirus* and *Embecovirus* have a lower CpG content than that of the *Nobecovirus* genomes.

To further confirm this pattern, we estimated the divergence times between SARS-CoV-2 and all other betacoronaviruses (Appendix A) and plotted them against their respective CpG contents. Figure 2A shows an increase in the CpG contents with the increase in the time of divergence of *Sarbecovirus*, *Hibecovirus,* and *Nobecovirus* genomes with SARS-CoV-2. However, this trend reverses due to the relatively low genomic CpG contents of *Merbecovirus* and *Embecovirus*. The overall correlation was not significant (*ρ* = 0.18, *p* = 0.12). However, after removing these two groups, the correlation becomes highly significant (*ρ* = 0.86, *p* < 0.00001) (Figure 2B). Looking at this result reveals that the CpG dinucleotides progressively declined in the SARS-CoV-2 lineage over time. For instance, the average CpG content of *Nobecovirus* genomes was 2.82%, which was almost two times higher than that estimated for SARS-CoV-2 and its close relatives (1.46%) (Figure 2B). Figure 1 (inset) and Figure 2 suggest that the systematic depletion of CpGs might have started in the SARS-CoV-2 lineage after the *Hibecovirus* + *Sarbecovirus* clade separated from *Nobecovirus* (see arrow—Figure 1).

### 3.2. Natural Selection and CpG Deficiency

The rate of CpG mutations is well known to be >10 times higher than that of the non-CpG nucleotides [29]. Therefore, the decline in the CpG contents could be explained by the process of mutation from CpG to non-CpG nucleotides. However, CpG nucleotides are present in functionally important positions of the genome as well. Therefore, mutation of those CpG dinucleotides will not be tolerated as they are required for the survival of the virus itself and, thus, are expected to be under natural selection [30,31]. To investigate the role of selection, we estimated the CpG contents of synonymous (neutral—under no selection) and nonsynonymous (under selection) sites of *Nobecovirus*, *Hibecovirus,* and *Sarbecovirus* genomes and plotted them against the times of divergence from SARS-CoV-2. Similar to the genomic correlation shown in Figure 2B, we observed highly significant correlations between the two variables for synonymous as well as nonsynonymous sites (Table 1). The slope of the regression line is almost four times much steeper for synonymous sites compared to that of nonsynonymous positions (Table 1). This is because the magnitude of CpG depletion was much higher for the former compared to the latter. For instance, the average CpG content of the synonymous sites of SARS-CoV-2 and its close relatives (deep blue in Figure 1) is 2.61 times higher than that estimated for *Nobecovirus* (red in Figure 1) (5.9% vs. 2.0%). In contrast, this difference was only 1.87 times for nonsynonymous sites (2.5% vs. 1.3%).

### 3.3. Evolution of Arginine Content and CpG

In order to further confirm the effects of natural selection, we examined the proportion of arginine content. This is because arginine is the only amino acid with codons containing CpG nucleotides in the first two positions (CGA, CGG, CGT, and CGC). However, arginine is also coded by two non-CpG codons (AGA and AGG) and, hence, two-thirds (4 out of 6) of the codons contain CpG dinucleotides in the first two positions. We first estimated the proportion of CpG containing the codons code for arginine in different betacoronavirus exomes and plotted this against the time of divergence from SARS-CoV-2. A significant positive correlation between the two variables was observed (Table 1). The CpG containing codons coded for 63.7% of the arginine in the exomes of *Nobecovirus* (Figure 3A), which is roughly as expected (2/3rd). Whereas this proportion dropped down to 42% in the exomes of SARS-CoV-2 and its close relatives owing to the loss of CpG dinucleotides in these genomes.

We then estimated the arginine content of the genomes of betacoronaviruses and plotted them against the time of divergence from SARS-CoV-2. This also revealed a significant positive correlation (Table 1). The average arginine content of SARS-CoV-2 and its close relatives was 3.5%, which was significantly less than that observed for *Nobecovirus* genomes (4.2%) (Figure 3B).

### 3.4. Observed and Expected Proportions of CpG (I_CpG_)

Although previous results clearly showed a systematic decline in the CpG content in the SARS-CoV-2 lineage over time, this could also be due to the reduction in GC contents in the genomes analyzed. To disentangle CpG content from GC content, we estimated the ratio of observed to expected GC content (*I_CpG_*) for the whole viral genomes using equation 2 (see methods). The correlation between *I_CpG_* and divergence times (Figure 4) was highly significant (*ρ* = 0.87, *p* < 0.00001) and very similar to that observed for the genomic CpG content (Figure 2B). The mean *I_CpG_* estimated for SARS-CoV-2 and its close relative genomes was 63% smaller than that observed for the *Nobecovirus* genomes (0.40 vs. 0.66). Similar analysis was performed, and highly significant correlations were observed for synonymous (*ρ* = 0.88, *p* < 0.00001) and nonsynonymous (*ρ* = 0.88, *p* < 0.00001) positions of the exomes of these viruses. The average *I_CpG_* estimated for nonsynonymous positions of *Nobecovirus* exomes was 0.92, which almost close to the expectation. However, this value was only 0.58 for the exomes of SARS-CoV-2 and its close relatives. On the other hand, the *I_CpG_* estimated for synonymous positions was 0.62 for the *Nobecovirus* exomes and only 0.37 for SARS-CoV-2 and its close relatives.

## 4. Discussion

Although a number of previous studies reported about the reduced CpG content of SARS-CoV-2 genomes [14,17,18,20], the present study showed that the CpG content has been progressively declining in this lineage over a longtime. We showed that this depletion might have started in the *Sarbecovirus* + *Hibecovirus* clade after it branched off from the *Nobecovirus* group (Figure 1). Based on the age of this node it is clear that the CpG content in SARS-CoV-2 lineage has been gradually declining over the past 1213 years. The cause for the reduction could be to evade the antiviral zinc finger protein of the hosts that targets CpG nucleotides [22,23]. Since the viral genomes belonging to *Sarbecovirus* + *Hibecovirus* groups infects many mammals, including human, bat, civet, pangolin, and mink (Appendix A), the progressive decline suggests that antiviral activities of zinc finger proteins appear to be similar across these mammals. Although CpG was much lower in the vast majority (>90%) of the SARS-CoV-2 genome, a few small pockets of the genome consisting of UTRs, envelop and nucleocapsid genes were shown to have relatively high CpGs [32]. Our analysis also showed a relatively higher CpG in these regions (3.5%) compared to that observed for the whole genome (2.1%). The combined CpG content of these regions (3.3%) is still lower for SARS-CoV-2 (and its close relatives) compared to *Hibecovirus* and *Nobecovirus* genomes (3.6%).

Our results showed that the reduction of CpG nucleotides occur in synonymous and nonsynonymous sites. The reduction in the CpG content of the latter suggests that not all nonsynonymous CpG nucleotides are under high selective constraints, and those under less or no selective constraints could have been mutated to non-CpG nucleotides. This is further confirmed by our analysis on arginine contents. We observed a shift in the codons coding for arginine amino acids from CpG to non-CpG codons in the SARS-CoV-2 lineage over time. However, we also observed a reduction in the proportion of arginine amino acids in this lineage, which suggests that some of these amino acid changes are tolerated in these viruses.

The results of the present study could be useful in designing universal vaccines [33]. For example, the reduction in CpG content suggests a higher rate of CpG→UpG mutations. Hence, we can predict that the CpG nucleotides in genomic regions (e.g., of the S gene) constituting the epitopes are more likely to be mutated to UpG compared to the mutations in the opposite direction. Therefore, knowledge about the most probable future mutation types is immensely useful in designing broad-spectrum epitopes that recognizes many strains of SARS-CoV-2, and its descendants emerging in the imminent future.

## 5. Conclusions

Using all representative genomes of the genus *Betacoronavirus*, we showed that the CpG dinucleotide has been gradually decreasing in the lineage of SARS-CoV-2 in the past 1213 years since it separated from *Nobecovirus*. Analyses on protein-coding exomes revealed that CpG reduction occurs at neutrally evolving—as well as selectively constrained—positions. Our results on the ratio of observed to expected CpG content clearly showed that the reduction in CpG content is not due to the reduction in the GC contents of the genome. These results are not only useful in understanding the evolution of SARS-CoV-2, but also in developing vaccines that are effective on current—as well as future—strains of COVID-19.

## Figures and Tables

**Figure 1 biology-10-00052-f001:**
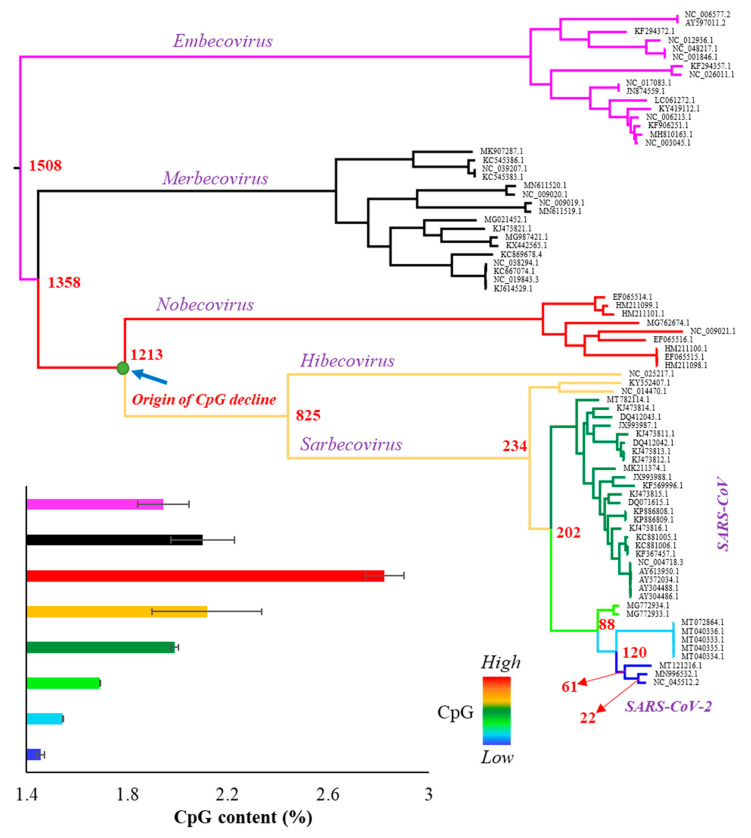
Phylogenetic relationship among 79 representative genomes (Appendix A) belonging to the genus *Betacoronavirus* including severe acute respiratory syndrome coronavirus 2 (SARS-CoV-2) (NC_045512.2). The tree was inferred using the maximum likelihood method. Red to blue colors indicate the levels of CpG content: red (2.82%), orange (2.12%), dark green (1.98%), pale green (1.69%), sky blue (1.54%), and dark blue (1.46%). Bayesian MCMC method base was used to estimate the time of divergence for each node that are shown in red (Appendix A). Inset: column graph showing the CpG contents (%) of various clades of the tree. Error bars indicate standard error of the mean.

**Figure 2 biology-10-00052-f002:**
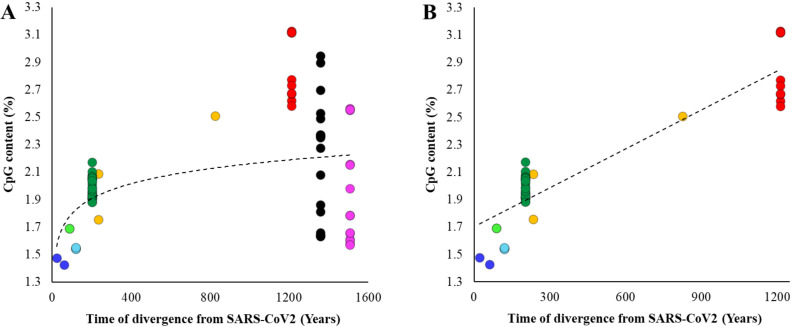
Relationship between divergence times of other betacoronaviruses from SARS-CoV-2 and their CpG contents. The divergence times for each node shown in Figure 1. (**A**) All genomes were included. The correlation is not significant (*ρ* = 0.18, *p* = 0.12). (**B**) *Embecovirus* and *Merbecovirus* genomes were excluded. The correlation is highly significant (*ρ* = 0.86, *p* < 0.00001). Best fitting regression lines are shown.

**Figure 3 biology-10-00052-f003:**
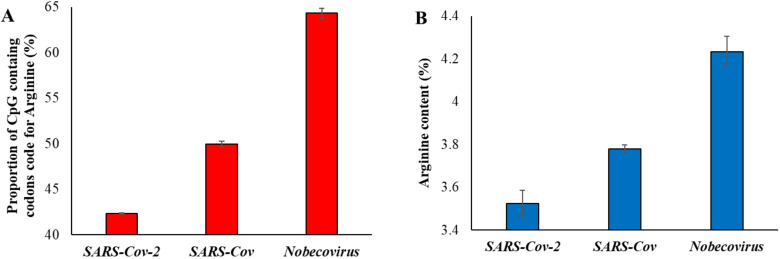
(**A**) Proportion of CpG containing the codons code for the amino acid arginine in three groups of virus genomes. (**B**) Proportion of arginine amino acid present in the exomes of SARS-CoV-2 (and its relatives—dark blue in Figure 1), SARS-CoV (dark green in Figure 1) and *Nobecovirus* (red in Figure 1). Error bars denote standard error of the mean.

**Figure 4 biology-10-00052-f004:**
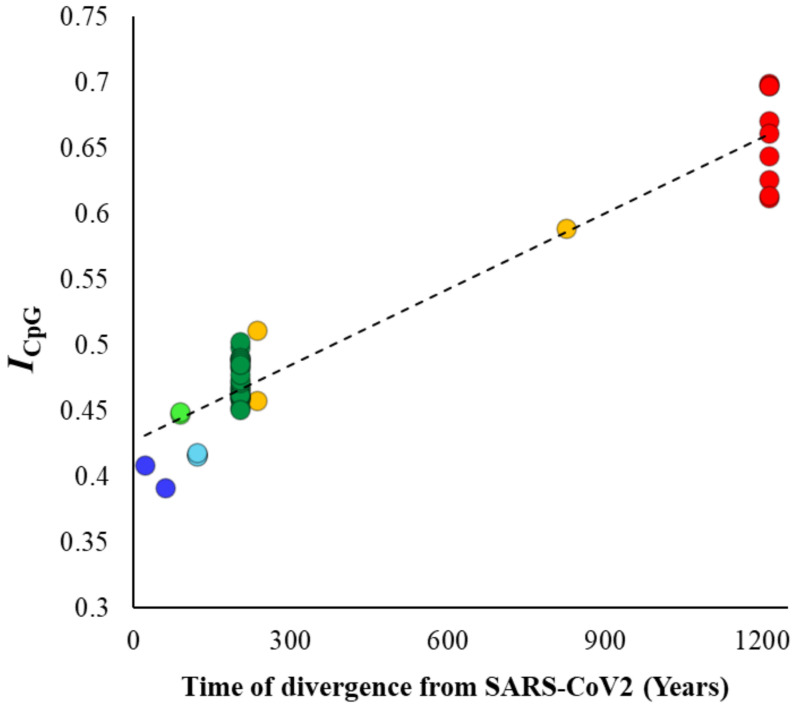
Correlation between the ratio of observed-to-expected CpG contents (*I_CpG_*—Equation (2)—see methods) of the betacoronavirus genomes and their respective times of divergence from SARS-CoV-2. The relationship is highly significant (*ρ* = 0.87, *p* < 0.00001).

**Table 1 biology-10-00052-t001:** Correlation coefficient and significance of relationships with the divergence times between SARS-CoV-2 and other betacoronaviruses.

*Y*-Axis	Slope of the Regression Line	Correlation Coefficient	Significance
Synonymous positions	0.0031	0.8	*P* < 0.00001
Nonsynonymous positions	0.0008	0.86	*P* < 0.00001
Arginine content	0.0005	0.79	*P* < 0.00001
Proportion of CpG containing codons coding for Arginine	0.0163	0.87	*P* < 0.00001

## Data Availability

No new data were created in this study. Data sharing is not applicable to this article.

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
