# Peer review of "The Long-Term Evolutionary History of Gradual Reduction of CpG Dinucleotides in the SARS-CoV-2 Lineage"

_biology, 2021, doi:10.3390/biology10010052_

Round 1
Reviewer 1 Report
In the presented manuscript Sankar Subramanian draws a maximum likelihood phylogenetic relationship between five genera of betacoronaviruses and suggests that low CpG frequencies observed in SARS-CoV-2 are a result of progressive evolution from the most common ancestor of genus Nobecovirus and Hibecovirus/Sarbecovirus.
This study indicates that betacoronaviruses showed a linear evolution in depletion of CpG content. I suspect that the correlation presented in this study is over simplistic, in the end authors studied sequences collected relatively recently and there is no direct timed relationship between the taxa.
Since this manuscript describes an interesting observation in terms of CpG frequencies and role in of natural selection in maintaining these frequencies, the manuscript could be re-written as an observation of CpG content among different genera of betacoronaviruses without the proposed correlation between CpG levels and divergence.
Minor corrections:
- SARS-CoV-2 is an RNA virus and does not produce DNA as part of its replication.
- Check ref 14, 15, 17 – 17 is Xia and he cites it a lot
- This reference is relevant and should be cited: Nchioua R, Kmiec D, Müller JA, Conzelmann C, Groß R, Swanson CM, Neil SJD, Stenger S, Sauter D, Münch J, Sparrer KMJ, Kirchhoff F. 2020. SARS-CoV-2 is restricted by zinc finger antiviral protein despite preadaptation to the low-CpG environment in humans. mBio 11:e01930-20. https://doi.org/10 .1128/mBio.01930-20
- Please note that authors cited above showed that certain parts of the genome have lower CpG frequencies than others.
- Ref 30 has not been peer reviewed.
Author Response
Reviewer 1
Comment: This study indicates that betacoronaviruses showed a linear evolution in depletion of CpG content. I suspect that the correlation presented in this study is over simplistic, in the end authors studied sequences collected relatively recently and there is no direct timed relationship between the taxa.
Since this manuscript describes an interesting observation in terms of CpG frequencies and role in of natural selection in maintaining these frequencies, the manuscript could be re-written as an observation of CpG content among different genera of betacoronaviruses without the proposed correlation between CpG levels and divergence.
Response: We agree with the reviewer that the pairwise distance correlation was simplistic and a direct timed relationship between taxa is important to support the observations of this study. To address this issue, we have conducted a sophisticated Bayesian MCMC analysis to estimate the actual times of divergence between the taxa. We have now used these divergence times to correlate with the CpG contents. This reanalyses also showed a highly significant correlation between split times and CpG contents. We have redrawn figures 2 and 4 as well as table 1. We rewrote parts of the main text and included a section on Bayesian time estimation. We have also provided the actual timetree in the supplementary material (Figure S1). The changes were highlighted in yellow (pages3-5 and 7).
Comment: SARS-CoV-2 is an RNA virus and does not produce DNA as part of its replication. Check ref 14, 15, 17 – 17 is Xia and he cites it a lot.
Response: The reviewer is correct SARS-CoV-2 is an RNA virus and does not produce DNA and we have checked the references mentioned by the reviewer regarding this. We have corrected this in the main text (page 1, line 20 and page 2, line 48).
Comment: This reference is relevant and should be cited: Nchioua R, Kmiec D, Müller JA, Conzelmann C, Groß R, Swanson CM, Neil SJD, Stenger S, Sauter D, Münch J, Sparrer KMJ, Kirchhoff F. 2020. SARS-CoV-2 is restricted by zinc finger antiviral protein despite preadaptation to the low-CpG environment in humans. mBio 11:e01930-20. https://doi.org/10 .1128/mBio.01930-20. Please note that authors cited above showed that certain parts of the genome have lower CpG frequencies than others.
Response: We have cited and added this reference. We have also discussed about the CpG variation in different parts of the genome (page 7, line 244-250).
Comment: Ref 30 has not been peer reviewed.
Response: This reference has been removed.
Reviewer 2 Report
This manuscript is exceptionally well written, in which the author investigated the evolutionary history of CpG dinucleotide deficiency in the SARS-CoV-2 lineage belonging to genus Betacoronavirus. The finding of this study is another important piece in the puzzle of understanding SARS-CoV-2 and developing potential new antiviral treatments and vaccines.
Methods and Results sections are precisely and clearly written. Figure 1 beautifully depicts the snapshot of the phylogeny.
The discussion section could be improved by adding a paragraph on how this ongoing depletion of CpG dinucleotide can affect future antiviral treatments, and vaccine target selection and development.
Specific Comment:
Line 129: Please add - ”As shown in figure 1- inset and figure 2,”
Author Response
Reviewer 2
Comment: The discussion section could be improved by adding a paragraph on how this ongoing depletion of CpG dinucleotide can affect future antiviral treatments, and vaccine target selection and development.
Response: We have now included a paragraph to discuss about the potential implications of CpG depletion in developing vaccines (page 8, paragraph 2)
Comment: Line 129: Please add - ”As shown in figure 1- inset and figure 2,”
Response: We have corrected this (page 3, line l34)